# Primary Colorectal Tumor Displays Differential Genomic Expression Profiles Associated with Hepatic and Peritoneal Metastases

**DOI:** 10.3390/cancers15174418

**Published:** 2023-09-04

**Authors:** Maximiliano Gelli, Christophe Desterke, Mohamed Amine Bani, Valérie Boige, Charles Ferté, Peggy Dartigues, Bastien Job, Geraldine Perkins, Pierre Laurent-Puig, Diane Goéré, Jacques R. R. Mathieu, Jerome Cartry, Michel Ducreux, Fanny Jaulin

**Affiliations:** 1Université Paris-Saclay, Gustave Roussy, INSERM, Dynamique des Cellules Tumorales (U-1279), F-94805 Villejuif, France; maximiliano.gelli@gustaveroussy.fr (M.G.); diane.goere@aphp.fr (D.G.); jacques.mathieu@gustaveroussy.fr (J.R.R.M.); jerome.cartry@gustaveroussy.fr (J.C.); michel.ducreux@gustaveroussy.fr (M.D.); 2Gustave Roussy, Département de Anesthésie, Chirurgie et Interventionnel, F-94805 Villejuif, France; 3Université Paris Saclay, INSERM, Modèles de Cellules Souches Malignes et Thérapeutiques (UMR1310), F-94805 Villejuif, France; christophe.desterke@inserm.fr; 4Gustave Roussy, Département de Biologie et Pathologie Médicale, F-94805 Villejuif, France; mohamed-amine.bani@gustaveroussy.fr (M.A.B.); peggy.dartigues@gustaveroussy.fr (P.D.); 5Université Paris-Saclay, CNRS, Inserm, US23, UMS3655, F-94805 Villejuif, France; bastien.job@gustaveroussy.fr; 6Gustave Roussy, Département de Médecine Oncologique, F-94805 Villejuif, France; valerie.boige@gustaveroussy.fr (V.B.); charles.ferte@gustaveroussy.fr (C.F.); 7Institut du Cancer Paris CARPEM, AP-HP, AP-HP Centre, Department of Hepatogastroenterology and Digestive Oncology, Hôpital Européen Georges Pompidou, 20 Rue Leblanc, F-75015 Paris, France; geraldine.perkins@aphp.fr; 8Sorbonne Université, USPC, Université Paris Descartes, Université Paris Diderot, Centre de Recherche des Cordeliers, INSERM, CNRS, F-75005 Paris, France; pierre.laurent-puig@parisdescartes.fr

**Keywords:** colorectal cancer dissemination, peritoneal metastases, liver metastases, organotropism, mucinous colorectal cancer

## Abstract

**Simple Summary:**

Metastatic spread is the main prognostic factor in patients with colorectal cancer (CRC). We investigated the preferential metastatic spread of CRC toward two prevalent and clinically relevant metastatic sites, liver and peritoneum. By comparing the differential gene expression profile of primary tumors with isolated liver or peritoneal metastases, we identified a 61-gene signature associated with a specific metastatic route. Primary CRC tumors expressing the peritoneal signature were characterized by epithelial mesenchymal transition and apical epithelial junction activation but also an implication of stem cell signaling pathway. We identified specific clinico-pathological correlations and prognostic impacts of liver and peritoneal signatures in a TCGA dataset. As a future perspective, biomarkers identified in the primary tumor may contribute to improved risk stratification for individualized patient follow-up and to identify new therapeutic targets for a precision-based approach.

**Abstract:**

Background: Despite improvements in characterization of CRC heterogeneity, appropriate risk stratification tools are still lacking in clinical practice. This study aimed to elucidate the primary tumor transcriptomic signatures associated with distinct metastatic routes. Methods: Primary tumor specimens obtained from CRC patients with either isolated LM (CRC-Liver) or PM (CRC-Peritoneum) were analyzed by transcriptomic mRNA sequencing, gene set enrichment analyses (GSEA) and immunohistochemistry. We further assessed the clinico-pathological associations and prognostic value of our signature in the COAD-TCGA independent cohort. Results: We identified a significantly different distribution of Consensus Molecular Subtypes between CRC-Liver and CRC-peritoneum groups. A transcriptomic signature based on 61 genes discriminated between liver and peritoneal metastatic routes. GSEA showed a higher expression of immune response and epithelial invasion pathways in CRC-Peritoneum samples and activation of proliferation and metabolic pathways in CRC-Liver samples. The biological relevance of RNA-Seq results was validated by the immunohistochemical expression of three significantly differentially expressed genes (*ACE2, CLDN18* and *DUSP4*) in our signature. In silico analysis of the COAD-TCGA showed that the CRC-Peritoneum signature was associated with negative prognostic factors and poor overall and disease-free survivals. Conclusions: CRC primary tumors spreading to the liver and peritoneum display significantly different transcriptomic profiles. The implementation of this signature in clinical practice could contribute to identify new therapeutic targets for stage IV CRC and to define individualized follow-up programs in stage II-III CRC.

## 1. Introduction

Colorectal cancer (CRC) is the third most common cancer and the second-leading cause of cancer-related mortality worldwide [1,2]. Approximately two thirds of all CRCs are actually diagnosed at localized or regional (I to III) stage for which surgical resection, eventually combined to systemic chemotherapy, represents the standard option achieving a 5-year overall survival (OS), ranging from 92% (stage I) to 73% (stage III) [2]. However, up to 60% of patients could develop metachronous metastatic disease (stage IV) [3] after primary tumor resection during surveillance [4], which drastically affects prognosis, with 5-year OS of 14% [1].

Distant metastases involve the liver and peritoneum in about two thirds of patients, and as isolated metastatic disease in 40% [5,6]. Except for the 5% of patients with microsatellite instable (MSI) tumors who may benefit from first-line immunotherapy, the standard treatment of stage IV microsatellite stable (MSS) CRC is still based on the combination of different systemic chemotherapies (FOLFOX, FOLFIRI or FOLRINOX) and targeted agents (anti-*EGFR* or anti-*VEGF*) according to patient and tumor characteristics (molecular biology, tumor sidedness and pattern of metastatic spread) [7]. In case of oligometastatic disease, the prognosis is largely dictated by the metastatic site and the resectability with only a minority of patients eligible to potentially local (surgery or local ablation) curative treatments, either initially or after systemic therapy. Isolated pulmonary disease exhibits higher response rates to systemic chemotherapy and better survival among stage IV CRC patients [8,9]. Among the other metastatic sites, only selected patients with limited liver (LM) and peritoneal (PM) metastases are eligible to potentially curative treatments with similar outcomes [10]. However, in a majority of these patients, the diagnosis occurs at advanced stages with extended metastatic burden, multi-metastatic pattern and chemo-resistant disease, resulting in limited chances of cure. In the absence of large-scale personalized medicine programs in CRC, surveillance or treatment intensification strategies represent, in all likelihood, should represent promising approaches. Unfortunately, intensive surveillance programs and prophylactic strategies based on surgical or medical treatments in high-risk patients defined according to specific clinico-pathological risk factors failed to improve disease-free survival (DFS) in randomized controlled trial studies [11,12,13,14]. More recently, circulating tumor DNA has been proposed as biomarker to detect the minimal residual disease after the curative resection of primary tumor [15,16]. Unfortunately, circulating tumor DNA levels are heterogeneous among the metastatic sites and particularly low for PM [17]. Nonetheless, up to now the underlying mechanisms of metastatic spread and organotropism of CRC remain ill-defined [18,19,20]. Gene expression profiling has been proposed as a valid approach for a wide range of applications in cancer research. In CRC, consensus molecular subtypes (CMS) is now recognized as a robust classifier with specific clinical and biological features [21], but its clinical impact still remains limited. DNA microarray techniques have been previously used in CRC to identify specific signatures within primary tumors, associated with metastatic potential [22]. Several studies investigated the differential gene expression (DGE) of CRC at different stages [23] or distant metastatic sites (nodes, liver and peritoneum) [23,24]. However, only a few studies [19,25] assessed the correlation between DGE of the primary tumor and the risk of dissemination toward distant organs.

In an attempt to elucidate the relevance of DGE involved in the metastatic dissemination of CRC, the present study aimed to analyze the molecular signature of CRC primary tumor associated with liver and peritoneal metastatic routes. Furthermore, in order to assess the clinical implications of these transcriptomic profiles, we further correlate these specific signatures with clinico-pathological characteristics and patients’ prognosis throughout an external in silico analysis within a large TCGA dataset.

## 2. Materials and Methods

### 2.1. Patients’ and Tissue Selection

Frozen samples from primary tumors obtained from CRC patients treated at Gustave Roussy cancer center and European Hospital Georges Pompidou were analyzed. Potentially resectable disease was defined according to metastatic site, tumor response to systemic chemotherapy and possibility of achieving a macroscopic resection with curative intent. Unresectable disease included a definitively unresectable metastatic disease, or presenting a progression after at least one line of systemic therapy enrolled in the MOSCATO-01 [26] (NCT01566019), a clinical trial aiming to evaluate the clinical benefit of targeted therapy guided by high-throughput genomic profiling.

Samples were prospectively collected and archived in both Centers during the study period and subsequently analyzed at Gustave Roussy Cancer Campus, Villejuif, France. The tumor tissue with highest cellularity on frozen sections was selected by the local pathologist. Tumor cellularity exceeding 30% was required for both primary tumor and metastatic tissues. Study data were collected and managed using REDCap electronic data capture tools hosted at Gustave Roussy Cancer Campus [27,28].

The surgical specimens were classified according to the TNM Classification (8th edition UICC/AJCC—2017). Clinico-pathological characteristics as well as the corresponding formalin-fixed paraffin-embedded (FFPE) tissue were collected for further analysis. Patients presenting a lack of data or specimen to correlate clinical data and gene expression profile, or presenting with co-malignancies, were excluded.

A specific written consent was obtained from all included patients. Dataset and specimens were irreversibly anonymized and researchers were blinded to the clinical and transcriptomic data during the analysis phase. The study was performed according to the recommendations of the local ethics committee of Gustave Roussy Cancer Campus, granted by Local Authorities, and was conducted in accordance with the Declaration of Helsinki.

### 2.2. RNAseq Transcriptome

Library preparation, capture and sequencing analysis were performed by IntegraGen, Evry, France. Libraries were prepared using the TruSeq Stranded mRNA kit (Illumina) following the manufacturer’s instruction. Briefly, the TruSeq stranded mRNA sample prep kit converts the poly-A containing mRNA in total RNA (1000 ng engaged in the process) into a cDNA library using poly-T oligo-attached magnetic bead selection. Following mRNA purification, the RNA was chemically fragmented prior to reverse transcription and cDNA generation. The fragmentation step resulted in an RNAseq library that included inserts ranging in size from approximately 400 m. The cDNA fragments then went through an end repair process, the addition of a single ‘A’ base to the 3′ end and then ligation of the adapters. Finally, the products were purified and enriched with PCR to create the final double stranded cDNA library, which was then purified and quantified by QPCR. Each transcriptome library was sequenced on an Illumina NextSeq 500 as paired-end 75 base pair reads. RNAseq base calling was performed using the Real-Time Analysis software sequence pipeline (RTA2) with default parameters. Sequence reads were mapped to the human genome build (hg19) using TopHat version 2.1.0, including Bowtie2 aligner [29]. Read counts were voom transformed with voom function in limma R-package version 3.48.3.

### 2.3. CRC Tumor Classification

RNA-Seq matrix normalized and “voom” transformed tumor samples were classified according to the Consensus Molecular Subtypes (CMS) classification from the CRC Subtyping Consortium. CMS assignment for each sample was performed using the CMSclassifier R package obtained from Synapse Project 2,623,706 (https://www.synapse.org/) and was applied with the RF (random forest) and SSP (single sample predictor) methods. This algorithm was downloaded at the following address: https://github.com/Sage-Bionetworks/CMSclassifier (accessed on 14 July 2023). A verification of concordance was done between the predictions of the two algorithms. Significance of CMS classes was assessed by Chi square test in accordance with the phenotypes of the tumors.

### 2.4. Bioinformatics Analyses

Bioinformatics analyses were performed in R software environment version 4.1.0 (18 May 2021) [30]. Principal component analysis (PCA) plots were drawn with autoplot function from ggfortify R-package version 0.4.12. Differential Expressed Gene (DEG) between tumor samples with implementation of False Discovery Rate correction (FDR) was performed with limma R-package version 3.48.3 [31]. Unsupervised clusterings were performed in R with Euclidean distances and complete method as options. Volcano plots display each gene’s −log10 (*p*-value) and log2 fold change with the selected covariate. Horizontal lines indicate False Discovery Rate (FDR) adjusted *p*-value thresholds. The expression heatmap was performed with pheatmap R-package version 1.0.12. Gene Set Enrichment Analysis (GSEA) was performed with standalone software version 4.1.0 on the Molecular Signature Database (MSigDb) version 7.4 [32] with Hallmark sub-library. Main significant gene sets were ranked based on their normalized enrichment scores (NES). After enrichment, the gene list were submitted for functional enrichments in ToppGene suite online application [33] in order to organize the significant gene sets in distinct classes of function.

### 2.5. Immunohistochemestry Assay

Immunohistochemical staining was performed on 4-μm thick unstained sections generated from FFPE blocks of the primary tumors on Ventana Automated Immunostainer (BenchMarker Ultra, Ventana Medical System, Inc., Tucson, AZ, USA) according to the manufacturer’s manual.

Anti-ACE2 (Cell Signaling, clone E5O6J, Ref #92485S, dilution 1:200), anti-DUSP4 (Proteintech, clone polyAb, Ref 66349-1-Ig, dilution 1:500), anti-CLDN18 (Proteintech, clone McAb, Ref21126-1-AP, dilution 1:400), and anti-CD45 (DAKO, clone 2B11+PD7/26, Ref M0701, dilution 1:250), rabbit or mouse IgG isotype antibodies were used for the immunostainings. The avidin/biotin-complex kit and 3,3-Diaminobenzidine (DAB) peroxidase substrate kit were used to develop the staining.

Finally, all the samples were counterstained with hematoxylin and mounted.

Appropriate positive and negative controls were run simultaneously for all antibodies tested.

### 2.6. Digital Image Analysis

All slides were scanned at 20X using VS200 scanner (Olympus, Tokyo, Japan). CD45 staining quantification was performed using QuPath v0.4.0 software (Bankhead, P. et al. QuPath: Open source software for digital pathology image analysis. Scientific Reports (2017)). Regions of interest were manually delineated by a trained pathologist. Inside these regions, tissue was automatically detected using a trained classifier. Color deconvolution (Hematoxylin, DAB and residual) was then applied to the images. Cells were detected based on their optical density, and positivity to CD45 was assessed based on the amount of DAB signal in each cell. Results are expressed as a number of stained cells by square millimeters of analyzed tissue.

Regions of interest were manually delineated by a trained pathologist (MAB). Inside these regions, tissue was automatically detected using a trained classifier. Color deconvolution (Hematoxylin and DAB) was then applied to the images. Cells were detected using Halo AI classifier. Each color component was then thresholded to determine the phenotype of each cell. Results are expressed as a number of stained cells, for each phenotype, by square millimeters of analyzed tissue.

### 2.7. TCGA CRC Firehose Cohort Validation

In order to explore clinical and molecular implications of CRC-liver and CRC-peritoneum signatures, we used the “TCGA Legacy Firehose” dataset, including 640 CRC tumor samples from 636 patients. RNAseq Zscore V2 standardization of RNAseq data, GISTIC copy number and patients and samples clinical information were integrated through Cbioportal web application [34]. After integration, 382 CRC tumor samples with transcriptomic, genomic and clinical data from 374 CRC patients were analyzed after stratification according to DGE profile. Altered group and unaltered group of patients in TCGA cohort was defined according to the expression of genes from established signatures.

### 2.8. Xcell Immune Infiltration

Immune tumor infiltration was estimated using xCell R-package version 1.1.0 available at the address: https://github.com/dviraran/xCell (accessed on 13 February 2023). Significance of cell subtype enrichment in each group was determined by limma R-package version 3.52.4 after Bayes and False Discovery Rate corrections. Downstream graphical representation of these results was performed by volcanoplot.

### 2.9. Statistical Analysis

Continuous variables are expressed as mean ± standard deviation (SD) or median with ranges according to the Gaussian distribution. Categorical variables are expressed as counts and frequencies. Categorical and continuous variables were compared using the chi-square test (or Fisher test, as appropriate) and Student *t*-test (Mann-Whitney or ANOVA, as appropriate), respectively. All tests were 2-sided, with a level of significance set at *p* < 0.05. Probabilities of OS and DFS were predicted from the time of the curative intent surgery until death or disease recurrence, respectively. Conventional statistical analyses were performed with R++ 15.03 and SPSS version 23.0 (SPSS Inc., IBM, Chicago, IL, USA).

## 3. Results

### 3.1. Study Cohort

Among 61 CRC patients recruited in two tertiary cancer centers between December 2011 and March 2016, primary tumor samples from 18 patients with CRC with LM or PM were selected. Primary tumor samples were then classified according to distant metastatic site in CRC-Liver (*n* = 12) and CRC-Peritoneum (*n* = 6) group. RNAseq analysis was retrospectively applied. Median age at diagnosis was 54 years. Primary tumor resection after preoperative chemotherapy was performed in 62.5% of patients with 94.1% of T3-T4 stage and 64.7% of nodal invasion. In three patients, tumors harboured a *BRAF* mutation in CRC-Liver group. One patient in CRC-Peritoneum group harbored a mismatched-repair deficiency (dMMR) (Table 1). Mucinous histological subtype was significantly more frequent in the CRC-Peritoneum group compared to CRC-Liver group (50% vs. 0% *p* value 0.044). The number of chemotherapy lines before sampling was significantly higher in CRC-Peritoneum group compared to CRC-Liver group (*p* value 0.034). The curative-intent surgery rate was similar (100% and 83.3%) in CRC-Liver and CRC-Peritoneum group.

After a median follow up of 61.5 months, median OS from diagnosis was two-fold significantly higher in CRC-Liver group, with a median survival of 81.5 months, compared to 44.3 months in CRC-Peritoneum group (Log-rank *p* value 0.015), but did not significantly differ after curative-intent surgery (70.5 and 44 months for CRC-Liver and CRC-Peritoneum groups, respectively). Details of baseline, treatments and outcomes are summarized in Table 2.

The CMS classification was assessed in primary tumor samples. Among primary tumor of CRC-Liver group, nine samples were classified as CMS2 (canonical), two as CMS 3 (metabolic) and one as CMS4 (mesenchymal). In CRC-Peritoneum group, four samples were classified as CMS4 (mesenchymal), one as CMS3 (metabolic) and one as CMS 1 (Microsatellite Instability Immune) (Figure 1). Thus, the CMSclassifier algorithm predicted a significantly higher rate of CMS2 in CRC-Liver samples (75% vs. 0%, *p* value 0.009), as well as a significantly higher rate of CMS4 in CRC-peritoneum samples (66.7% vs. 8.3%, *p* value 0.033), showing different molecular traits in these primary tumors with LM and PM.

### 3.2. Differential Gene Expression between Primary Tumor Samples and Immunohistochemical Validation

DGE analyses between CRC-Liver and CRC-Peritoneum groups were carried out in attempt to assess the relevance of DGE involved in the metastatic route. Unsupervised gene filtration was performed in an RNA-sequencing experiment by removing univariate genes between samples. After this filtration (Figure 2a), size of the transcriptome was reduced to 18,413 variable genes. Unsupervised PCA performed with filtrated RNAseq accurately stratified CRC-Liver and CRC-Peritoneum primary tumor samples (Figure 2b).

These results show that near 15% of the transcriptome of these primary tumors is modulated based on their associated metastatic site. The volcano plot of differentially expressed genes between CRC-Liver and CRC-Peritoneum samples depicted in Figure 2c clearly confirms the results of PCA. Overall, 61 genes were significantly differentially expressed between CRC-Liver and CRC-Peritoneum samples. These results confirm that primary CRC tumors have distinct transcriptomic profiles depending on their final metastatic site.

The genes-fold change ranged from −7.30 to 5.75 (FDR *p* < 0.05). All significantly differentially expressed genes are presented in the heat map in Figure 2d allowing a good stratification in the unsupervised clustering of the dissemination process (peritoneal vs. liver metastatic spread). We selected three significantly differentially expressed genes in order to assess our RNAseq results in term of protein expression using immunohistochemistry. ACE2 was homogenously expressed in primary tumor samples from CRC-Liver group, while CLDN18 and DUSP4 were homogenously expressed in Peritoneum samples with a Log2 fold change of −3.267, 4.891 and 2.764 respectively (Appendix A). Immunohistochemical analysis revealed a significantly higher positive staining for ACE in CRC-Liver samples (90% vs. 10%, *p* value 0.0027) and DUSP4 in CRC-Peritomeum (83.3% vs. 10%, *p* value 0.0033). Similarly, the mean staining for CLDN18 was significantly higher in CRC-Peritoneum sample (*p* value 0.0005) (Figure 3). Median DFS and OS from curative surgery were two-fold, but not significantly, higher in patients with primary tumor harboring ACE2 (22.8 vs. 6.7 months *p* value 0.120 and 74.0 vs. 43.4 months *p* value 0.090, respectively). On the contrary, the expression of CLDN18 and DUSP4 in primary tumor samples was associated with worse outcomes with poorer OS for DUSP4 staining (37.4 vs. 65.0 months, *p* value 0.035) (Appendix A). These results further support the clinical and biological validation of transcriptomic analysis for these genes.

### 3.3. Differential Pathways’ Activation between CRC Primary Tumors with Liver and Peritoneal Spread

In order to characterize the underlying pathways associated with CRC-Liver and CRC-Peritoneum samples, a functional analysis based on the Hallmarks sub-library of MSigDb database was performed. Main significant gene sets were ranked based on their NES. After enrichment, the significant gene sets were organized in distinct classes of function.

The expression pattern of enriched gene sets in the CRC-Liver samples displays an important enrichment of transcription class function gene sets via *E2F* (elongation factor 2) target and *G2M* checkpoint gene set, followed by mitotic spindle and oxidative phosphorylation processes implicated in stress function. CRC-Liver samples also became enriched in metabolic functions, including cholesterol, fatty acid, bile acid and glycolysis pathways (Figure 4a and Appendix A).

In contrast, in CRC-Peritoneum group, the activation of immune response and epithelial invasion as epithelial mesenchymal transition (EMT) and apical junction pathways were observed among the top rank pathways. Furthermore, CRC-Peritoneum group showed a significant enrichment of gene sets implicated in stem cell function, such as Wnt/β-catenin signaling and hedgehog signaling (Figure 4b and Appendix A).

According to the immune signature observed in the CRC-Peritoneum group, we assessed the immune cells infiltration using CD45 staining. Semiautomatic analysis of unselected immune population showed a significantly higher immune infiltration in CRC-Peritoneum samples (Figure 5a,b). We also analyzed the immune enrichment by xCell algorithm, showing a significant myeloid infiltration in CRC-Peritoneum samples (Figure 5c). These results support the findings obtained by GSEA analysis.

Altogether, these results suggest that CRC primary tumors with PM present an important immune response with EMT and apical epithelial junction activation, but also with implication of stem cell signaling pathways, while primary tumors with LM present a proliferative and metabolic signature.

### 3.4. Prognostic Implication of CRC-Peritoneum and CRC-Liver Signatures in TCGA Database

In order to analyze the clinical implications of these transcriptomic signatures and their prognostic impact, we analyzed the clinico-pathological characteristics associated with CRC-Peritoneum and CRC-Liver signature using a large TCGA dataset from 374 CRC samples [35].

In the TCGA database, the CRC-Peritoneum signature was associated with a higher rate of lympho–vascular or vascular invasion (*p* value 0.004 and 0.02 respectively) and rectal and colon mucinous adenocarcinoma (*p* value 8.15 × 10^−4^, confirming our results in Table 2). This is in line with the OncoTree tumor classification system, confirming a higher rate of Mucinous Adenocarcinoma of the Colon and Rectum (MACR) sub-category (*p* value 0.037) (Appendix A). In the TCGA database, the presence of the CRC-Peritoneum signature was associated with poor prognosis in terms of DFS (*p* value = 0.042) and OS (*p* value 0.029) (Figure 6a,b). By contrast, CRC-Liver signature did not affect clinical prognosis in terms of OS (*p*-value 0.805) and of DFS (*p* value 0.660) as compared to the remnant patients in the TCGA cohort (Figure 6c,d).

## 4. Discussion

In this study, we aimed to identify the molecular differences between CRCs harboring either isolated liver or peritoneal metastatic disease. Resolving CRC organotropism is essential, since LM and PM should be eligible to curative onco-surgical strategies, particularly if detected in their early onset [36,37,38]. Unfortunately, current strategies still result in high recurrence rate [39,40] and poor chances of cure, as low as 16% at 10 years [41,42]. This highlights the need to integrate additional biomarkers to conventional mutational (*RAS*, *BRAF)* profiles and MMR status into modern treatment algorithms. Clinical and pathological characteristics of patients only partially explain the heterogeneity in term of metastatic spread and outcomes among patients with CRC. In this study, we observed a significantly higher rate of mucinous primary tumors in the CRC-peritoneum group, accounting for more than 50% of patients. The mucinous subtype is a well-established risk factor for peritoneal spread in the literature [43,44] and justifies a proactive approach in these patients. To date, randomized studies exploring different types of treatment intensification (i.e., proactive approach) failed to improve survivals in selected high-risk patients defined by clinical parameters [13,14], raising the limits of the actual risk stratification criteria and standard treatment protocols.

Our analyses identified a transcriptomic landscape of CRC translated into two distinct gene expression profiles based on a 61-gene signature associated with a specific metastatic spread toward the liver or the peritoneum. This correlation between specific gene sets’ enrichment and organotropism is based on a discovery cohort of 18 CRC patients. Despite the limited sample size, we confirmed the biological relevance of the transcriptomic signature using immunohistochemical staining of three significantly differentially expressed genes. Finally, we performed an in silico validation of our 61-gene signatures in the large TCGA dataset including over 374 patients [21]. The peritoneal signature disclosed specific pathological associations and poor prognosis in the TCGA dataset. Interestingly, the correlation with mucinous histology (blinded to the metastatic evolution) indirectly cross-validated our transcriptomic results.

RNA-sequencing and expression profiling data provide a comprehensive and accurate understanding of carcinogenesis and could be soon integrated in the patient risk stratification to improve cancer management. The CMS is a widely used transcriptome-based classification that encompasses meaningful biological features of CRC with clinical impact in terms of tumor response [45,46] and prognosis [47]. In the CRC-peritoneum group, we observed a significantly higher rate of CMS4 subtype (66.7%) in line with previous data reported in the literature [48]. Interestingly, the rate of CMS4 drastically increases across the clinical stages, reaching up to 50% of stage IV disease in mucinous CRC [49], which represent over 60% of CRCs with synchronous PM [48]. These data confirmed the higher metastatic potential of this CMS subtype, particularly in the peritoneal cavity. However, no specific applications of CMS classification are actually available in clinical practice, specially due to allocation issues [47] and intra-tumor heterogeneity [50].

Transcriptomic analysis of the CRC-liver and CRC-peritoneum group showed significantly DGE profiles. Pathway enrichment analysis identified interesting clues in metastatic spreads. Expected key drivers of cancer progression were found. The CRC-liver signature was characterized by proliferation/transcription and metabolic pathways while the CRC-peritoneum signature was associated with metastatic dissemination including the EMT, stem cell pathways and the inflammatory signaling mediated by IL6 and STAT-3. Chronic inflammation is thought to play an important role in the development and dissemination of cancer [51]. IL6, a proinflammatory cytokine produced in tumor-bearing states, promoted in CRC growth, EMT and metastatic spread [52,53] in association with dysfunctional antitumor immunity [54]. Multiple cell types in the tumor microenvironment produce IL-6, leading to activation of *JAK/STAT3* signaling in both tumor and tumor-infiltrating immune cells. Preclinical studies aiming to modulate the expression and/or function of *STAT3* have demonstrated important roles of *STAT3* in the progression of multiple cancers, including CRC [55]. *STAT3* also promotes resistance to conventional chemotherapy and targeted therapies [56], and such is the case in the 75% of patients with PM (*RAS* mutated) [57]. Direct targeting of IL-6, such as Tocilizumab [58] have received FDA approval for various malignancies, and other novel inhibitors of the *IL-6/JAK/STAT3* signaling pathway are currently in clinical and/or preclinical development. Interestingly, we also observed an activation of the stemness gene set function in CRC-peritoneum samples. This is in line with the study by Barriuso et al. comparing primary tumors and paired-PM with Nanostring technique that identifies local inflammation and ‘stemness’ activation as key drivers of CRC cells colonization of the peritoneal cavity [59]. Finally, the transcriptomic analysis also yielded EMT, a driver of stemness and cancer progression [60].

The concept of organotropism has been previously validated in other malignant epithelial tumors (i.e., breast and pancreatic cancers) [61,62], as the result of a nonrandom multifactorial process. Proteins implicated in EMT such as epithelial plasticity and cellular junctions are major candidates in this process [18]. Moreover, the biological processes implicated in different metastatic routes are highly independent. Recent data support the potential role of additional physiological factors, such as age and sex, in the metastatic dissemination of CRC [63,64]. In hypermethylated CRC, we identified a metastatic cascade toward the peritoneum through the production of a tumor sphere with inverted polarity [65,66]. Unlike dMMR CRCs, these tumor intermediates display specific properties enabling their invasion and dissemination in tissues with low adhesion, such as lymphatic and peritoneal environments [67,68]. Several studies have reported the utility of gene expression profile of primary tumor to predict the later metastatic spread across different types of cancers [20,69], including CRC [70,71]. Transcriptomic analyses were used to explore the intra-tumor heterogeneity between matched primary and metastatic tumors [59,72] or among different metastatic sites [23], but only few of them directly analyze in detail the metastatic dissemination process within primary tumor tissue. In CRC, Jacob et al. [73] identified an activation of the Toll-like receptor cascade in liver metastatic spread, but failed to find a specific gene signature of peritoneal dissemination most likely due to the limited number of genes resulting from the NanoString technique. Network analysis identified significant upregulation of immune-related and EMT pathways in high-risk patients [74], providing a potential tool for patient risk stratification. Our results confirmed that the metabolic pathways downregulation favors a high immune and stromal infiltration in the microenvironment and is associated with poor prognosis CRC phenotype [75]. Additionally, we were able to confirm in CRC-Peritoneum samples the immune enrichment using immunostaining techniques and xCell algorithm.

The application of our peritoneal signature to the TCGA database aimed to assess its prognostic impact in a large dataset. Due to an absence of data concerning the pattern of metastatic sites in the TCGA cohort, this analysis was not able to test the prediction of peritoneal spread of the signature. Nevertheless, we could confirmed the association with mucinous histological subtype, a well-established clinical entity with poor prognosis [76]. Thus, these data indirectly support the correlation between the signature and the peritoneal spread of this type of tumor.

Overall survival of CRC patients with early stages has significantly increased in the last decades, as a result of improved staging procedures, refined surgical techniques and perioperative radio- and chemo-therapy regimens. Nevertheless, limited progress has been observed in patients with metastatic disease. Metastatic spread of CRC remains a main health-care issue accounting for 15% of all cancer-related deaths world-wide [2]. Colorectal cancer is classically known as a sequential organ-specific dissemination process [77]. More recently, new evidence has shown that the activation of the metastatic potential is already encoded in the primary tumor at early stages, rather than subsequently acquired by clonal expansions of an ancestral fraction of tumor cells with metastatic potential [20,69,78]. The achievement of the metastatic dissemination cascade is dependent from acquired genetic mutations but also from epigenetic events and interactions with the microenvironment [79]. Accumulating evidence has supported the interaction between local tumor microenvironment and somatic mutations, epigenetic modulation, and microbial components as crucial in the tumor response to immunotherapy [80]. The rapid evolution of single cell RNA-seq technologies has accelerated the identification of diverse cellular populations and phenotypic states within different tumor samples. An extensive analysis of the immune components across primary and metastatic site, as recently shown by Zhang et al. in 17 CRC patients, will be crucial for establishing the link between distinct tumor types and their interplay with different immune features of a specific organ. These techniques, integrating molecular profile of primary tumors, will open opportunities for developing tailored strategies in selected CRC patients [81].

Nowadays, considering the potentially curative-intent of surgery for LM and PM, these types of approach may anticipate, or even prevent, the diagnosis of limited hepatic or peritoneal spread. Identification of patients at increased risk of isolated liver- or peritoneal-relapse would allow adjuvant specific therapies targeting the hepatic parenchyma or peritoneal cavity. Proactive approaches based on loco-regional treatment intensification (intra-hepatic or peritoneal perioperative chemotherapy) have been previously investigated to reduce recurrence risk in high-risk clinical-defined patients with heterogeneous results [13,82]. Our transcriptomic signature could participate in the refinement of selection criteria beyond clinical parameters to select “high-risk” patients for peritoneal or liver relapse, a key point to drive a patient-tailored and evidence-based approach in the management of CRC. Although the sample size could seem small, only a few studies with limited number (<20) of patients have recently been published in this topic [25,45]. Due to the limited sample size, a further validation in a larger cohort should be required, but the absence of information regarding the metastatic site in large CRC dataset precluded this possibility. This points out the persistent gap between molecular and clinical settings, the latter being mandatory for medical applications. This is not the case for breast cancer, where recent studies analyzed in detail the molecular subtype-based organotropism of different metastatic sites [83]. According to the chance of cure for mCRC, these aspects should presumably be taken into account in the next collaborative CRC consortiums to accelerate the introduction of tailored approaches in cancer research. Up to now, transcriptomic-guided immunostaining techniques still represent a valid option.

## 5. Conclusions

In conclusion, the present study shows that peritoneal and hepatic metastatic spread could be identified by simple 61-gene profiling of primary tumors. In the short term, this could outline a solid risk-stratification algorithm for patients with early CRC and could provide valuable insights for “multi-molecular” based personalized therapies and prognosis prediction.

## Figures and Tables

**Figure 1 cancers-15-04418-f001:**
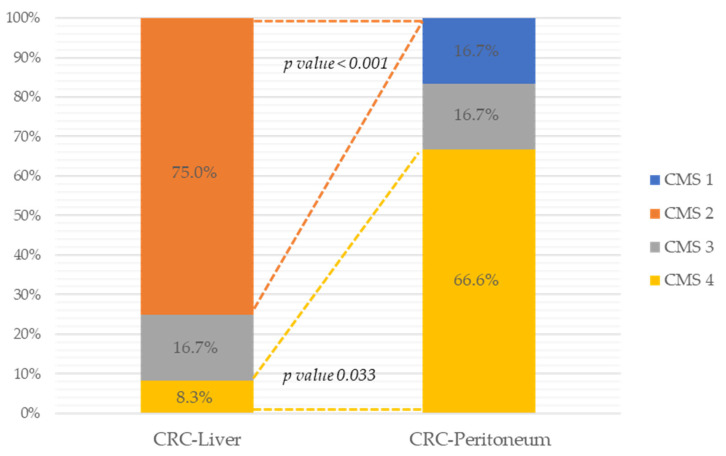
CMS classification of 18 primary CRC stratified according to CRC-Liver and CRC-Peritoneum group.

**Figure 2 cancers-15-04418-f002:**
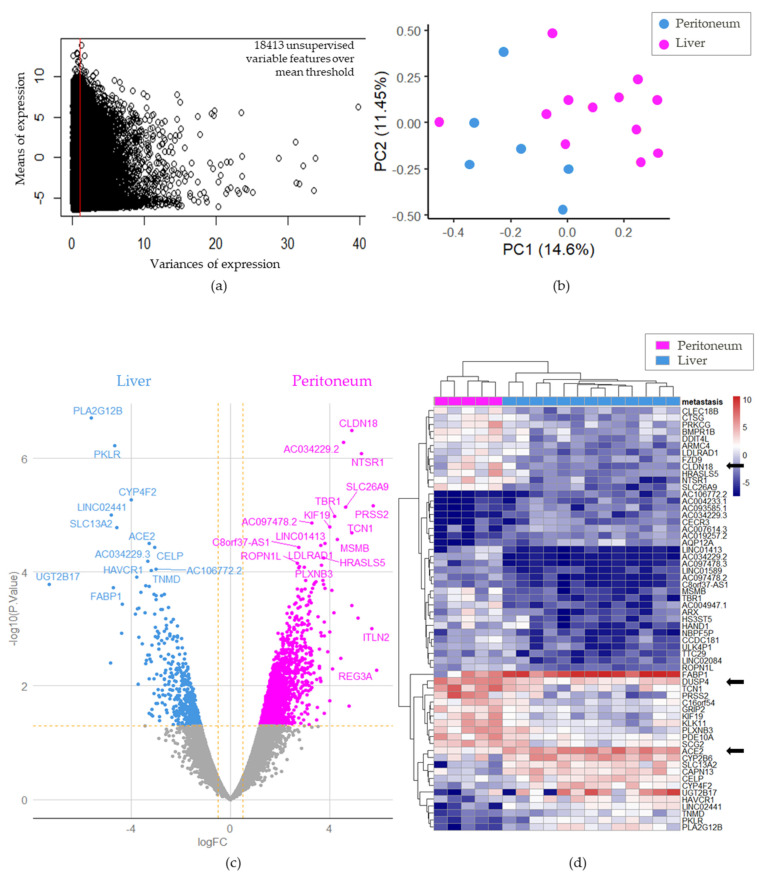
Primary tumor DGE between CRC−Liver and CRC−Peritoneum groups: (**a**) Filtration of invariable genes in selected RNAseq (12 CRC primary tumors with liver metastasis and 6 CRC primary tumors with peritoneum metastasis); (**b**) unsupervised PCA on whole variable transcriptome stratified according to the metastatic spread; (**c**) Volcano−plot of differential expressed genes between CRC−Liver (blue) and CRC−Peritoneum (pink) groups; (**d**) Expression heatmap of differential expressed genes between CRC−Liver (blue) and CRC−Peritoneum (pink) groups. Black arrows display genes selected for immuno−histochemistry assays.

**Figure 3 cancers-15-04418-f003:**
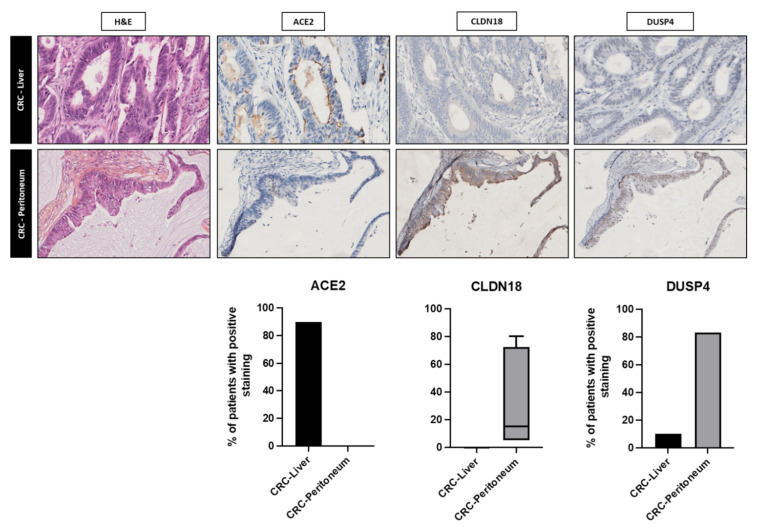
Immunohistochemical evaluation of the three most relevant proteins from the RNAseq results. The upper panel (×200 magnification) represents a CRC liver with an apical ACE2 staining in tumor cells and no expression of CLDN18 and DUSP4. The bottom panel (×200 magnification) is a CRC peritoneum sample with cytoplasmic expression of CLDN18 and a nuclear expression of DUSP4. Chart bars indicates the corresponding percentage of patients with positive staining (ACE2 and DUSP4) and percentage of cells with positive CLDN18 staining for in CRC−Liver and CRC-Peritoneum groups.

**Figure 4 cancers-15-04418-f004:**
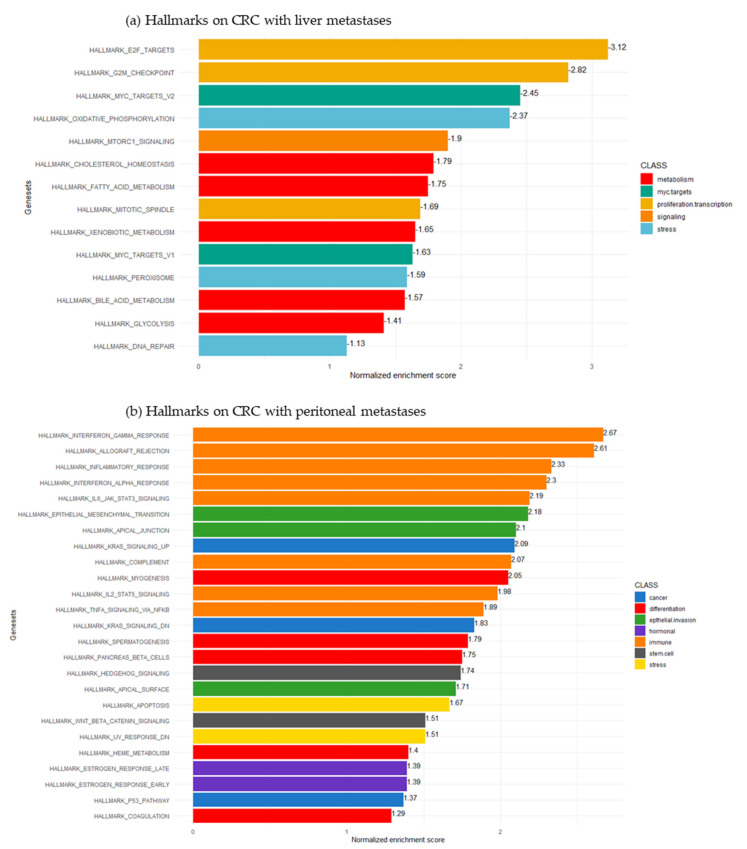
Bar−plot of normalized enrichment score of hallmarks obtained by gene set enrichment analysis: (**a**) CRC primary tumors with LM harbored an intense proliferation and transcription associated to activation of metabolism; (**b**) CRC primary tumors with PM harbored an epithelial remodeling associated with activation of inflammation and stem cell pathways.

**Figure 5 cancers-15-04418-f005:**
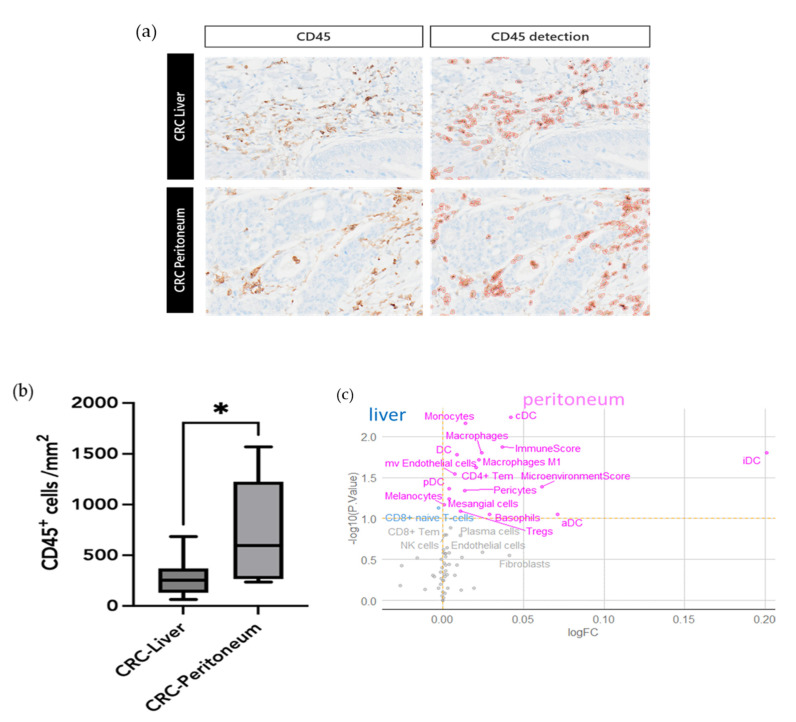
Tumor immune infiltration in CRC-Peritoneum samples: (**a**) CD45 immunohistochemistry images (×200 magnification) of CRC-Liver and CRC-peritoneum samples. CD45 staining was quantified using digital pathology detection; (**b**) Box plot representing the number of CD45 stained cells detected per 1mm^2^; (**c**) Immune enrichment of primary tumor samples between CRC Liver and CRC Peritoneum groups. iDC: immature dendritic cells, aDC: activated dendritic cells, Tregs: T regulatory cells, pDC: plasmacytoid dendritic cell, cDC: conventional dendritic cells, Tem: effector memory T cell, mv Endothelial cells: microvascular endothelial cells, NK cells: Natural killer cells. * *p* value < 0.05.

**Figure 6 cancers-15-04418-f006:**
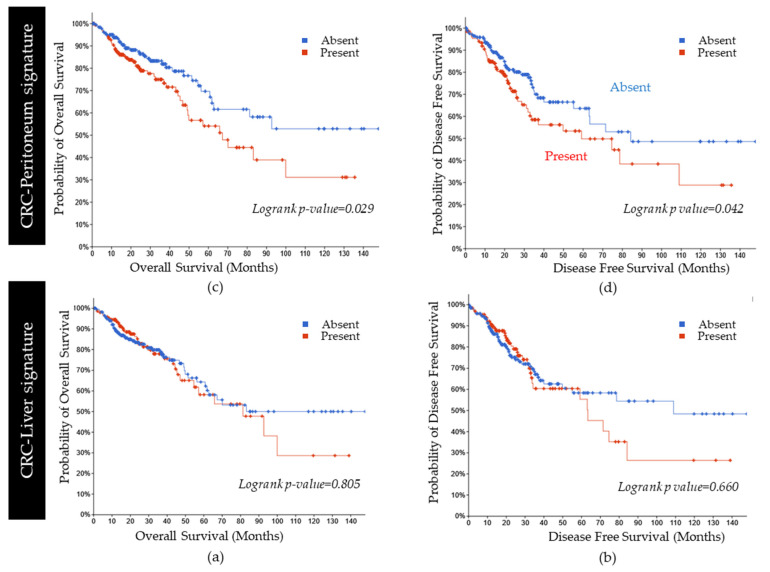
Kaplan–Meier survival curves according to CRC-Liver and CRC-Peritoneum groups: Kaplan–Meier and log-rank test on Overall (**a**) and Disease Free (**b**) survival stratified according to presence or absence of the CRC-Peritoneum signature (TCGA Firehose). Kaplan–Meier and log rank test on Overall (**c**) and Disease Free (**d**) survival stratified according to presence or absence of the CRC-Liver signature (TCGA Firehose).

**Table 1 cancers-15-04418-t001:** Baseline and treatments characteristics of the analyzed patients.

ID	Cohort	Gender	Center	Sidedness	Primary Tumor Resection	T	N	Type	WHO Subtype (% muc)	*RAS* Status	*BRAF* Status	MMR Status	Regimen	Targeted Therapy	RECIST	Surg	DFS
1T	CRC-Liver	M	1	Left sided	Yes	T1/T2	N1	Sync	NOS (0%)	wt	NA	NA	None	None	-	Yes	22.8
2T	CRC-Liver	F	1	Left sided	Yes	T3/T4	N2	Sync	NOS (0%)	wt	mut	MSS	None	None	-	Yes	86.5
4T	CRC-Liver	F	1	Right sided	Yes	T3/T4	N2	Met	NOS (1–25%)	wt	NA	MSS	None	None	-	Yes	3.0
5T	CRC-Liver	F	1	Right sided	Yes	T3/T4	N1	Sync	NOS (0%)	wt	mut	MSS	None	None	-	Yes	8.8
6T	CRC-Liver	M	1	Left sided	Yes	T3/T4	N1	Met	NOS (0%)	wt	wt	MSS	Ir	anti-VEGF	SD	Yes	136.0
7T	CRC-Liver	M	1	Left sided	Yes	T3/T4	N2	Sync	NOS (0%)	mut	wt	NA	Ox	anti-VEGF	PR	Yes	7.5
8T	CRC-Peritoneum	M	1	Left sided	Yes	T3/T4	N2	Sync	NOS (1–25%)	mut	wt	MSI	Ox	anti-VEGF	PR	Yes	6.7
9T	CRC-Liver	M	1	Left sided	Yes	T3/T4	N0	Sync	NOS (0%)	mut	wt	MSS	Ox	anti-VEGF	PR	Yes	106.0
10T	CRC-Liver	F	1	Right sided	Yes	T3/T4	N2	Sync	NOS (0%)	mut	wt	NA	None	None	-	Yes	4.1
12T	CRC-Liver	M	1	Right sided	Yes	T3/T4	N1	Sync	NOS (0%)	mut	wt	MSS	None	None	-	Yes	5.8
13T	CRC-Peritoneum	F	1	Left sided	Yes	T3/T4	N0	Sync	NOS (26–50%)	wt	wt	MSS	Ox	None	-	Yes	6.5
15T	CRC-Liver	M	1	Left sided	Yes	T3/T4	N1	Sync	NOS (0%)	mut	wt	MSS	Ox	anti-VEGF	PR	Yes	4.3
16T	CRC-Peritoneum	F	1	Left sided	Yes	T3/T4	N0	Sync	Muc (>50%)	wt	wt	MSS	Ir	None	PR	Yes	9.3
19T	CRC-Peritoneum	M	1	Left sided	Yes	T3/T4	N0	Sync	Muc (>50%)	mut	wt	MSS	Ox	anti-VEGF	SD	Yes	7.6
20T	CRC-Peritoneum	F	1	Left sided	Yes	T3/T4	N0	Sync	Muc (>50%)	mut	wt	MSS	Ox	anti-VEGF	-	Yes	13.3
21	CRC-Peritoneum	M	1	Left sided	None	-	-	Sync	NOS (0%)	mut	wt	MSS	Ox	anti-VEGF	PR	None	NA
62	CRC-Liver	M	2	Left sided	Yes	T3/T4	N0	Sync	NOS (0%)	wt	wt	MSS	NA	NA	-	NA	NA
63	CRC-Liver	M	2	Left sided	Yes	T3/T4	N2	Sync	NOS (0%)	wt	mut	MSS	NA	NA	-	NA	NA

Gender: F, female, M, Male; Center: 1 Gustave Roussy, 2 HEGP; Type: Sync, synchronous, Met, metachronous; WHO, World Health Organization, Muc, mucinous subtype (%) indicates percentage of mucinous component; RAS: wt, wild-type, mut, mutated; MMR, Mismatch Repair: MSS, microsatellite stable, MSI microsatellite instable; Regimen: Ox, oxaliplatin, Ir, Irinotecan; RECIST: PR: Partial response, SD: Stable disease according RECIST criteria; DFS: Disease free survival from curative surgery; NA: not available.

**Table 2 cancers-15-04418-t002:** Baseline and treatment characteristics.

		Overall	CRC-Liver	CRC-Peritoneum	*p* Value
Age at diagnosis	Median (Q1, Q3)	54.0 (43.25, 65.25)	53.0 (43, 64.25)	55.5 (6.25, 64)	0.85
Gender	Female	7 (38.89%)	4 (33.3%)	3 (50.0%)	0.864
	Male	11 (61.11%)	8 (66.7%)	3 (50.0%)	
Center	GR	16 (88.89%)	10.0 (83.3%)	6 (100%)	0.791
	HEGP	2 (11.11%)	2 (16.7%)	0 (0%)	
Sidedness	Left-sided	14 (77.78%)	8 (66.7%)	6 (100%)	0.316
	Right-sided	4 (22.22%)	4 (33.3%)	0 (0%)	
T stage	T1/T2	1 (5.88%)	1 (8.33%)	0 (0%)	1
	T3/T4	16 (94.12%)	11.0 (91.7%)	5 (100%)	
N stage	N0	6 (35.29%)	2 (16.7%)	4 (80.0%)	0.053
	N1-2	11 (64.71%)	10.0 (83.3%)	1 (20.0%)	
Histological subtype	NOS	15 (83.33%)	12 (100%)	3 (50.0%)	0.044
	Mucinous	3 (16.67%)	0 (0%)	3 (50.0%)	
Intravascular emboli	None	8 (61.54%)	4 (44.4%)	4 (100%)	0.20
	Yes	5 (38.46%)	5 (55.6%)	0 (0%)	
Perineural invasion	None	3 (50%)	2 (50.0%)	1 (50.0%)	1
	Yes	3 (50%)	2 (50.0%)	1 (50.0%)	
*RAS* status	Mutated	9 (50%)	5 (41.7%)	4 (66.7%)	0.617
	Non mutated	9 (50%)	7 (58.3%)	2 (33.3%)	
*BRAF* status	Mutated	3 (18.75%)	3 (30.0%)	0 (0%)	0.408
	Non mutated	13 (81.25%)	7 (70.0%)	6 (100%)	
*TP53* status	Mutated	7 (50%)	5 (55.6%)	2 (40.0%)	1
	Non mutated	7 (50%)	4 (44.4%)	3 (60.0%)	
Mismatch repair system	MSI	1 (7.14%)	0 (0%)	1 (20.0%)	0.76
	MSS	13 (92.86%)	9 (100%)	4 (80.0%)	
Chemotherapy before sampling	None	6 (37.50%)	6 (60.0%)	0 (0%)	0.06
	Yes	10 (62.50%)	4 (40.0%)	6 (100%)	
Number of line before sampling	0	6 (37.50%)	6 (60.0%)	0 (0%)	0.034
	1	7 (43.75%)	4 (40.0%)	3 (50.0%)	
	2	2 (12.50%)	0 (0%)	2 (33.3%)	
	3 or more	1 (6.25%)	0 (0%)	1 (16.7%)	
Chemotherapy protocol Line 1	None	6 (37.50%)	6 (60.0%)	0 (0%)	0.058
	Oxaliplatin IV	6 (37.50%)	2 (20.0%)	4 (66.7%)	
	Irinotecan IV	3 (18.75%)	1 (10.0%)	2 (33.3%)	
	Oxaliplatin IA	1 (6.25%)	1 (10.0%)	0 (0%)	
Targeted therapy Line 1	None	8 (50%)	6 (60.0%)	2 (33.3%)	0.607
	Bevacizumab	8 (50%)	4 (40.0%)	4 (66.7%)	
Number of cures	0	6 (37.50%)	6 (60.0%)	0 (0%)	0.019
	1 to 4	1 (6.25%)	0 (0%)	1 (16.7%)	
	5 to 7	7 (43.75%)	2 (20.0%)	5 (83.3%)	
	8 or more	2 (12.50%)	2 (20.0%)	0 (0%)	
Response Line 1	Objective response	6 (75%)	3 (75.0%)	3 (75.0%)	1
	Stable	2 (25%)	1 (25.0%)	1 (25.0%)	
Curative R0 surgery	None	1 (6.25%)	0 (0%)	1.00 (16.7%)	0.792
	Yes	15 (93.75%)	10.0 (100%)	5.00 (83.3%)	
Recurrence	None	4 (30.77%)	4 (40.0%)	0 (0%)	0.546
	Yes	9 (69.23%)	6 (60.0%)	3 (100%)	
Death	None	8 (44.44%)	5 (41.7%)	3 (50.0%)	0.502
	Yes	10 (55.56%)	7 (58.3%)	3 (50.0%)	

## Data Availability

The data presented in this study are available in this article (and Appendix A).

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
