# Peer review of "Primary Colorectal Tumor Displays Differential Genomic Expression Profiles Associated with Hepatic and Peritoneal Metastases"

_cancers, 2023, doi:10.3390/cancers15174418_

Round 1
Reviewer 1 Report (New Reviewer)
Prediction of metastasis can be important for selection of optimal therapy. Topic of manuscript is very suitable for cancer a have only few points.
In the baseline characteristics of the entire cohort is mentioned age patients, have this parameter any influence on the risk of metastasis, or the prognosis?
Rasa of patients can be significantly effect on the prediction, it was monitored?
Minor
In the manuscript, are unapproved/non-implemented repairs.
Template is from last years.
Author Response
Comments and Suggestions for Authors
Prediction of metastasis can be important for selection of optimal therapy. Topic of manuscript is very suitable for cancer a have only few points.
We sincerely appreciate the attention devoted to our study by the Referee and we are grateful for the valuable insights and comments provided. We are pleased to learn that the overall assessment of the Referee reflects the relevance of this topic and the objective of our study.
We recognize the significance of a rigorous review process and are eager to follow your suggestions to further refine our article.
- In the baseline characteristics of the entire cohort is mentioned age patients, have this parameter any influence on the risk of metastasis, or the prognosis?
We thank the referee for the opportunity to clarify this point. Age has been classically reported as a key prognostic factor in digestive cancers since several decades. Higher age ranges are classically associated with a higher risk of toxicity during systemic therapy as well as a higher morbidity in the setting of curative surgery. Some additional aspects of age and senescence beyond the conventional physiological features have been recently reported.
Firstly, recent data from adjuvant trials in stage III CRC showed unvaforable tumor biology and worse oncological outcomes in younger patients despite more aggressive treatments. Secondly, recent data (PMID: 35104771 DOI: 10.1016/j.canep.2022.102112, PMID: 36765761 DOI:10.3390/cancers15030803) further highlight the role of age-related changes in tumor biology including modifications in tumor microenvironment, immune response and tumour biology. As results, despite a higher risk of CRC in older patients, a reduction of metastatic spread of CRC in patients higher than 70 years old have been shown. In our study, baseline characteristics were presented in Table 2. Median age at CRC diagnosis was not significantly different between CRC-Liver and Peritoneum groups (53.0 vs 55.5, p-value=0.85). Only mucinous subtype of primary tumours was significantly associated with peritoneal spread at univariate analysis. However, different metastatic patterns have been recently described across increasing age groups (PMID: 36765761 DOI: 10.3390/cancers15030803). Accordingly, Perotti et al. showed a higher rate of mucinous histology in specific age ranges. Altogether, these results could influence the specific metastatic pattern of CRC in the different ages of patients. These data further emphasis that organotropism in CRC is driven by different multifactorial molecular events. Secondly, the incidence of sporadic CRC in patients under the age of 50 years is increasing worldwide and associated with poor prognosis. The young onset CRC has been recently identified as an emerging public health issue by the international societies and will represent a new field of research in this domain. Consequently, depicting the biological and environmental factors, as well as their interactions, that affect the characteristics and clinical behaviour of CRC in homogenous group of patients with single metastatic sites represent the first step to decipher tumour heterogeneity of CRC.
According to these data, we implemented some reference in the manuscript in this regard.
- Rasa of patients can be significantly effect on the prediction, it was monitored?
RAS mutations play a crucial role in tumorigenesis with a strong predictive value. Beyond the no benefit of the anti-EGFR therapy in stage-IV CRC, a specific pattern of metastatic spread related to RAS status has been widely debated in the literature. Besides of the predictive role after resection of CRLM in term of relapse, RAS mutation has been associated with specific patterns of recurrence (lung, bone and brain)(Kim et al., 2012; Bonnot and Passot, 2019), although the causal effect is difficult to validate through retrospective studies. Instead, the negative synergic effect of the co-occurrence of somatic mutations (RAS and TP53, APC and PIK3CA) seems more evident(Chun et al., 2019). In our cohort, RAS, RAF, TP53 and microsatellites stability status were presented in Table 2. We identified a higher rate of RAS mut CRC in CRC-Peritoneum. This result is not statistically significant in our cohort, but in accordance with the fact that right sided CRC are both enriched in RAS mutation and peritoneal metastases(Loree et al., 2018; Passot et al., 2018).
Minor
- In the manuscript, are unapproved/non-implemented repairs. Template is from last years.
We are grateful to the referee for raising this issue related to the previous revision. In the current version, we have taken into account all changes validated by the co-authors.
Concerning the template, we agree about this misleading. According to the Editorial office instructions indicated in the mail’s Editor Board we currently amended the version available in the section “Manuscript for revision” according the latest Word template available online (https://www.mdpi.com/files/word-templates/cancers-template.dot).
Reviewer 2 Report (New Reviewer)
The article entitled “Primary colorectal tumor displays differential genomic expression profiles associated with hepatic and peritoneal metastases” by Gelli et al., shows very interesting and promising results to predict metastases in CRC patients. The article is well written and introduced; however, some points must be amended to improve the quality of the manuscript.
- Firstly, the article should be read and approved by all authors. The erased lines in the introduction denotes that it has not been properly reviewed
- Since it is a translational research that could impact on the future treatment of patients with a different pattern of metastases, I miss in the introduction a paragraph describing the systemic treatment of metastatic CRC. Surgical approaches and combinational treatments.
- I cannot understand why authors considered the cut-off point to include patient samples as low as 10% of cellularity. Here, RNA from normal tissue may have. Please justify this decision.
- Please include the approval number of the Ethical Review Board of your institution in this research.
- Please include in statistical analysis section the algorithm used for the estimation of the minimum sample size required for the study.
- Please review the whole manuscript and correct style of genes.
- A validation of the molecular signature for overall and disease-free survival at protein level by IHC of tumor samples from CRC patients must be included in the study.
Some clarity and usage of English has to provided in the manuscript.
Author Response
Comments and Suggestions for Authors
The article entitled “Primary colorectal tumor displays differential genomic expression profiles associated with hepatic and peritoneal metastases” by Gelli et al., shows very interesting and promising results to predict metastases in CRC patients. The article is well written and introduced; however, some points must be amended to improve the quality of the manuscript.
We extend our sincere gratitude to Reviewer 2 for the attention dedicated to our study submitted to “Cancers.” We also wish to express our appreciation for recognizing the significance of the subject matter we have explored. The thoughtful consideration and valuable feedback are deeply valued.
We appreciate the opportunity to respond to your specific comments point by point and address every aspect highlighted in your report, which will undoubtedly refine our article’s clarity and precision. We are confident that this revision will contribute to advancing the quality of our manuscript.
- Firstly, the article should be read and approved by all authors. The erased lines in the introduction denotes that it has not been properly reviewed
We are grateful to the Reviewer for this correction, and we apologized for this disregard. We confirm that the previous version has been validated by all co-authors but some revisions were not duly accepted. We have corrected these discrepancies.
- Since it is a translational research that could impact on the future treatment of patients with a different pattern of metastases, I miss in the introduction a paragraph describing the systemic treatment of metastatic CRC. Surgical approaches and combinational treatments.
We are extremely grateful to the Reviewer for the attention toward to clinical management of stage IV CRC. Unlike other cancers, among stage IV CRC patients the benefit of immunotherapy is limited to a <5% of tumours with microsatellite instability (MSI) and tumours harbouring “targetable” oncogenic drivers with FDA approval is even rarer (<3%). Therefore, during the last decades, the number of effective systemic regimens in CRC increased as the result of combination of the existing drugs (doublet and triplet chemotherapy) with targeted agents. In order to follow this recommendation, the standard of care of stage IV CRC according to the recent guidelines(Cervantes et al., 2022; Yoshino et al., 2023) are briefly integrated in a specific paragraph of the Introduction section of the revised manuscript.
- I cannot understand why authors considered the cut-off point to include patient samples as low as 10% of cellularity. Here, RNA from normal tissue may have. Please justify this decision.
We are grateful for your meticulous review of our manuscript. We apologized for this mistake related to previous version of the manuscript (including a larger population). According to MOSCATO-01 (MOlecular Screening for Cancer Treatment Optimization) protocol previously published(Massard et al., 2017; Koeppel et al., 2018), tumor cellularity was assessed by a senior pathologist on a haematoxylin and eosin slide and only tumour samples presenting more than 30% (and not 10% as previously indicated) of tumor cells were considered as contributive for RNASeq analysis in order to exclude samples without enough tumour cells in microenvironment. We acknowledge the misunderstanding you have identified about the potential impact on the quality of the technique. Consequently, we amended the section 2.1 indicating the 30% cut-off instead 10% previously mentioned. We sincerely apologize for this oversight on our part.
- Please include the approval number of the Ethical Review Board of your institution in this research.
The protocol was approved by the “Comité de protection des personnes” Ile de France III n° Am6752-5-2914 in 09/08/2015 and is now mentioned in the dedicated section of the manuscript (‘Institutional Review Board Statement’).
- Please include in statistical analysis section the algorithm used for the estimation of the minimum sample size required for the study.
In RNA-sequencing experimental map it is not possible to use an algorithm to determine sample size because during the analyses more than 30 thousand hypothesis (rather than only one) are done corresponding to an hypothesis for each quantified gene which has been tested to be differentially expressed between CRC-Liver and CRC-Peritoneum samples. During DEG analysis a False discovery Rate (FDR) correction was performed on p-values of differential genes in order to minimized false positive errors accumulated on hypothesis of each tested genes. Assessment of a minimal FDR corrected q-value allowed to retain gene which were found significantly representatives of differential expression between the two CRC samples. If the list of significant genes is consequent after FDR correction, the number of samples was statistically sufficient to explain their heterogeneity. Limma R-package algorithm was used to perform this statistical adjustment and is now clearly described in the section 2.4 page 12 in the current version of the manuscript.
- Please review the whole manuscript and correct style of genes.
We agree about the misleading about the style used for human gene in the manuscript. We provided the necessary corrections in the current version of the manuscript according to the international guidelines (human genes in italic style and capital letters, protein in capital letters) to enhance the accuracy and quality of our work.
- A validation of the molecular signature for overall and disease-free survival at protein level by IHC of tumor samples from CRC patients must be included in the study.
We are grateful to the Reviewer to point out this interesting insight. We assessed the expression of 3 proteins associated with our signature by immunohistochemical staining but we omitted to explore the potential correlations with clinical outcomes.
Then, we analysed disease free survival and overall survival according to the expression of these proteins in the entire cohort of patients. Results and KM survival curves are presented in the Table below.
|
|
ACE2 |
|
CLDN18 |
|
DUSP4 |
|
|
|
Pos (9) vs Neg (6) |
p value |
Pos (5) vs Neg (10) |
p value |
Pos (5) vs Neg (10) |
p value |
|
DFS |
22.8 vs 6.7 |
.120 |
7.6 vs 7.5 |
.347 |
7.6 vs 7.5 |
.281 |
|
OS from surgery |
74.0 vs 43.4 |
.090 |
43.4 vs 65.0 |
.092 |
37.4 vs 65.0 |
.035 |
|
OS from diagnosis |
80.9 vs 52.1 |
.106 |
52.3 vs 79.7 |
.238 |
41.3 vs 79.7 |
.059 |
Median disease free and overall survivals from curative surgery were two-fold, but not significantly, higher in patients with primary tumour harbouring ACE2 (22.8 vs 6.7 months and 74.0 vs 43.4 months respectively). On the contrary, expression of CLDN18 and DUSP4 in primary tumour sample was associated with worse outcomes reaching statistically significance in term overall survival for DUSP4 staining. These data, despite the limited statistically significance mainly related to the limited sample size, confirm our previous results in term of association between these markers and specific metastatic patterns (Liver and Peritoneum).
We propose, according to the Reviewer’s comment to include these results as supplementary materials in the section 3.2 “Differential gene expression between primary tumor samples and immunohistochemical validation”.
Comments on the Quality of English Language
Some clarity and usage of English has to provided in the manuscript.
We diligently worked toward improving the language and ensuring greater clarity throughout the manuscript by contribution of a native English speaker.

Round 2
Reviewer 2 Report (New Reviewer)
Thanks for the amended version of the manuscript.
The version provided by authors has been significantly improved
This manuscript is a resubmission of an earlier submission. The following is a list of the peer review reports and author responses from that submission.
Round 1
Reviewer 1 Report
The paper from Gelli et. al, reports the transcriptomic analysis of CRC metastatic patients and focuses on the differences between primary tumors bearing peritoneal or liver metastasis. Differentially expressed genes and pathways are identified using in silico studies, which might have some prognostic value. While the topic is of interest to the field, the manuscript lacks the validation and the statistical robustness, as well as enough novelty to warrant publication in the Cancers journal.
Major issues:
1. It is unclear why only 18 samples out of the 77 available have been analyzed. The study would strongly benefit from an increased number of analyzed samples; n=6 and n=12 for CRC-Peritoneum and CRC-Liver, respectively, do not constitute a sufficient number to draw robust and biologically meaningful conclusions, in particular, because some of the samples are derived from patients who received previous treatment, while others did not. It is unclear why only primary tumors are analyzed when multiple metastasis samples are available.
2. The first classification of C1 and C2 is confusing, and the biological meaning is not clear. The contribution of the stroma is reported without data to support it; then liver and peritoneal metastasis are chosen for further analysis with a rather undefined rationale.
3. While the analysis of the transcriptional profile of liver vs. peritoneal mets does indeed cluster nicely, the sample numbers are too small to draw any conclusion without further biological validation. For example, the authors claim increased immune activation or EMT signatures, but these claims must be validated at least with an ancillary technique (e.g., IHC) and on a broader sample size. Functional validation in cellular and/or mouse models would, of course, be a “smoking gun”.
4. The meaning of the TCGA survival data is unclear: the definition of altered and unaltered gene signature is obscure.
5. The association between mucinous CRC and peritoneal metastasis has already been described, and it does not constitute a novel element. Importantly, the mucinous histological subtype, which is predominant in the metastatic peritoneal tumors, might explain most of the transcriptional differences reported by the authors.
Minor points:
- Long tables in the main text are not reader-friendly
- The paper would benefit from the editing by a scientific editor, as sentences are often difficult to decipher.
Reviewer 2 Report
The manuscript entitled “Primary colorectal tumor displays differential genomic expression profiles associated with hepatic and peritoneal metastases” describes an interesting idea of the use of RNA-seq to find a gene expression panel of genes that would allow to predict metastatic properties of primary colorectal tumor samples. The authors identified 61 differentially-expressed genes between hepatic and peritoneal metastases of primary colorectal cancer, and the verification of this gene panel in colorectal cancer patients may help to choose appropriate treatment regimen. With the continued decline in sequencing costs and identification of novel biomarkers, such analyses are bringing us closer to precise medicine. Therefore, the presented analysis is interesting and important. However, the heterogeneity of cancers even in the same tumor groups, may impede the results conclusions. Also, we have to remember that changes in transcriptome of cancer cells are frequently transient and regulated epigenetically. These two problems should be discussed in more detail in the manuscript.
Also, several other minor issues warrants my attention:
- The authors retrospectively used frozen clinical samples to run RNA-seq analyses and to identify 61 genes that were differentially expressed between CRC-liver and CRC-peritoneum cohort. Since, assuming the bioavailability of frozen samples, the validation of expression of several most differentiated genes in both cohorts should be performed with qPCR assays. The results then should be compared with the NGS data.
- Supplementary Table 3 in section Chemotherapy before sampling lacks yes/no options.
- Although the manuscript is well written, it contains several grammar errors or typos that need to be corrected.
Reviewer 3 Report
Gelli et al. performed transcriptomic analysis on 77 tumors and assessed the clinic-pathological association of the current cohort and the COAD-TCGA cohort. They found the differences of CMS distribution between CRC-liver and CRC-peritoneum groups. They also used TCGA datasets to validate that CRC-peritoneum signature is associated with higher mutation and worse prognosis. Overall, it is descriptive research focused on the transcriptomic analysis and the sample size is not outstanding.
Recently, Zhang et al. (PMID: 35303421) published a similar article and it was based on single-cell technology. This makes this article less novel. Moreover, the whole analysis was based on transcriptomic data alone, while the mutation data are very important as well. I wonder whether authors could use their own datasets for the mutation association analysis and the prognosis analysis, instead of using the TCGA dataset, even if the validation is important as well. Besides, the immune profiles can easily be acquired through transcriptomic analysis, I also wonder why the authors did not perform analysis like MCP-counter, Cibersort etc for better understanding of the immune information from those samples.
Finally, the 61-gene signature is not practical. 61 genes are required to be sequenced in order to identify the potential clinical outcomes. It is obvious that if more samples are used or sequenced in this study could decrease the necessary number of the gene signatures, indicating that the current signature is not optimized.